Neglected Tropical Diseases

# A randomised controlled trial of raw honey for the healing of ulcers in leprosy in Nigeria

Paul Alumbugu Tsaku[1]*, Sunday Odihiri Udo[1], Pius Sunday[1], Anthony Meka[2], Linda Chinonso Ugwu[2], Abiola Oladejo[3], Joshua Akinyemi[3], Akinyinka Omigbodun[3], Sopna Choudhury[4], Jo Sartori[4], Onaedo Ilozumba[4], Samuel Watson[4], Richard Lilford[4]

1 The Leprosy Mission Nigeria, Fort Royal Estate, Abuja, Nigeria, 2 RedAid-Nigeria, Enugu, Nigeria, 3 Clinical Trials Unit, University of Ibadan Teaching Hospital, Ibadan, Nigeria, 4 Institute of Applied Health Research, College of Medical and Dental Sciences, University of Birmingham, Edgbaston, Birmingham, United Kingdom

* tsaku_pa@yahoo.com, tsakup@tlmnigeria.org

## Abstract

### Introduction

Chronic neuropathic ulcers remain a debilitating complication of leprosy, with limited evidence for effective treatments. Honey has been recommended to promote wound healing in other chronic ulcers. However, its efficacy in ulcer healing has not been rigorously evaluated.

### Methods

This dual-centre, prospective, single-blinded, randomised controlled trial compared raw honey dressings (n = 65) versus standard saline dressings (n = 65) for leprosy-associated foot ulcers in Nigeria. Participants with ulcers (2–20 cm$^2$, ≥ 6 weeks duration) were randomised 1:1, stratified by ulcer size. Primary outcomes were complete healing by 84 days and healing rate, assessed through blinded digital planimetry. Secondary outcomes included ulcer recurrence and/or new ulcer development at 6 months. A total of 130 participants were randomised in the study.

### Results

Complete healing occurred in 29.2% of honey-treated ulcers versus 24.6% with saline (adjusted HR 1.26, 95% CI 0.64-2.47). At 6 months, recurrence rates were similar (honey 13.5% vs saline 10.2%). The honey group showed a non-significant trend toward faster healing (p = 0.076). No treatment-related adverse events occurred.

### Conclusion

While honey dressings showed a modest advantage in healing rate, the difference was not statistically significant. The results suggest honey may be a safe, culturally

**Data availability statement:** All the data obtained and/or analysed in the study are publicly available in the following sites: https://www.isrctn.com/ISRCTN10093277, https://doi.org/10.1101/2023.07.14.23292603 and https://www.birmingham.ac.uk/research/applied-health/research/nihr-right-leprosy.

**Funding:** This research was funded by the National Institute for Health and Care Research (NIHR) (NIHR200132 to PT, SO, POS, AM, LU, AO, JA, AO, SC, JS, OI, SW, and RL) using UK aid from the UK Government to support global health research. NIHR ARC West Midlands supports RL. The views expressed in this publication are those of the author(s) and not necessarily those of the NIHR or the UK Department of Health and Social Care. The funders had no role in the study design, data collection and analysis, the decision to publish, or the preparation of the manuscript. All authors received a salary from the funder- NIHR, UK.

**Competing interests:** The authors have declared that no competing interests exist.

acceptable option in resource-limited settings. This study provides high-quality data for inclusion in future systematic reviews.

## Trial registration

ISRCTN10093277. Registered on 22 December 2021.

## Author summary

Leprosy causes neuropathic ulcers. These ulcers are often hard to heal, especially when they appear on the side of the foot where they are subjected to ongoing pressure. It has been suggested that honey applications may promote ulcer healing. This study tested whether raw, local honey (known for its natural antibacterial and healing properties) could do better than standard saline dressings. We enrolled 130 people with chronic foot ulcers, randomly allocated in a 1:1 ratio to either receive twice-weekly applications of honey or standard saline dressings. We tracked healing over 84 days using photographs measured by blinded experts. There was no statistical difference in the rate of healing across intervention and control groups, albeit with a slightly faster rate of healing with honey. By day 84, 29% of honey-treated ulcers had fully healed, compared to 25% with saline, but again, the result did not cross the conventional threshold. There were no treatment-related complications in either group. The overall improvement rate was slow, suggesting that the ulcers in this study were 'hard-to-heal'. While our study offers scant support for honey treatment, it does not prove that it is ineffective.

## Introduction

In leprosy, ulcers result from repeated trauma due to nerve impairment and sensory loss [1,2]. Nerve impairment results from both the direct effects of the causative organism, *Mycobacterium leprae* (*M. leprae*) complex in the peripheral nerves, and the host's immune response to it, leading to inflammation and demyelination in peripheral nerves [3,4]. As with diabetes and other causes of neuropathy, the combination of loss of sensation and deformities leads to chronic, recalcitrant, and recurring ulcers. It is estimated that 30 – 50% of people affected by leprosy eventually develop foot ulcers [5,6].

There has been recent advocacy for high-quality research on ulcer treatment and prevention in leprosy, specifically advocating for randomised controlled trials (RCTs) [7]. Being a Neglected Tropical Disease (NTD), there is a paucity of documentation on the management of leprosy, especially on the treatment of ulcers. Traditionally, the management of leprosy ulcers involves the regular soaking of the affected area in water and the use of paraffin oil for debridement and to promote self-healing. This practice constitutes what is termed 'self-care' in leprosy [8,9]. Additional methods to

promote healing include leukocyte and platelet-rich fibrin (L-PRF) [6], biomaterials [10], and removable offloading devices, but none of these have been proven to be effective [11,12].

In this study, we conducted a randomised controlled trial of raw African honey for the healing of ulcers in leprosy. The use of honey as a therapeutic agent in the treatment of wounds is an ancient practice, with the earliest documented report recorded in the Edwin Smith papyrus (2600 – 2200 BCE). Honey is a viscous, supersaturated solution containing sugars, water, amino acids, vitamins, minerals, enzymes, and many other substances derived from nectar gathered and modified by the honeybee, *Apis mellifera* [13,14]. Studies have shown that honey promotes wound healing, stimulates tissue growth, facilitates debridement and epithelization, reduces odurs, reduces oedema and exudates, and possesses antimicrobial properties [15–17].

Currently, there is a lack of scientific evidence on the use of honey for the treatment of leprosy ulcers. A recent Cochrane systematic review has reported several studies with unclear outcomes for trials with honey on venous leg ulcers, diabetic foot ulcers, and mixed chronic wounds [7]. The review concludes that most of the reported studies were of low quality with a high risk of bias due to non-blinding of observers, and statistical heterogeneity was evident across studies.

## Methods

### Ethics statement

The study protocol was approved by the National Health Research Ethics Committee in Abuja (Ref: FHREC/2022/01/09/04-02-22); the University of Nigeria Teaching Hospital in South East Nigeria (Ref: UNTH/HREC/2023/01/504); and the University of Birmingham (Ref: ERN_21–1388). Informed consent was obtained from all participants following a detailed explanation of the study procedures. Each participant received an information sheet and was given at least 24 hours to consider their involvement before providing written consent. Additionally, indemnity insurance was secured for all participants to ensure coverage in the event of any serious adverse occurrences related to study participation.

### Trial design

This was a dual-centre, prospective, single-blinded, parallel-group, 1:1 individually randomised controlled trial. The study duration was 29 months (March 2022 – August 2024), including post-discharge follow-up at 6 months after randomisation. The trial was registered on the International Standard Randomised Controlled Trial Number (ISRCTN) registry, with the number: ISRCTN10093277 (https://www.isrctn.com/ Registered on 22 December 2021). The study protocol has also been published [18].

The study pathway is shown in Fig 1.

### Study setting

The study centres are The Leprosy Referral Hospital, Chanchaga, Minna, Niger State, in North-central Nigeria, and St. Benedict Tuberculosis and Leprosy Relief Hospital, Ogoja, Cross River State, in South-south Nigeria. The Leprosy Referral Hospital, Chanchaga is a specialist hospital operated by the government of Niger state with support from The Leprosy Mission Nigeria, while the St Benedict Tuberculosis and Leprosy Relief Hospital, Ogoja is a TB and leprosy referral hospital located in Ogoja, a town in the northern part of Cross River State of Nigeria, owned by the Catholic church in Cross River State, Nigeria, with the technical support from Red-Aid Nigeria formerly the German Leprosy and TB Relief Association (DAHW). The sites were selected for their specialization in leprosy care, established partnerships with the research team (The Leprosy Mission Nigeria and RedAid-Nigeria), and geographic diversity (north-central vs. south-south Nigeria) to enhance generalisability.

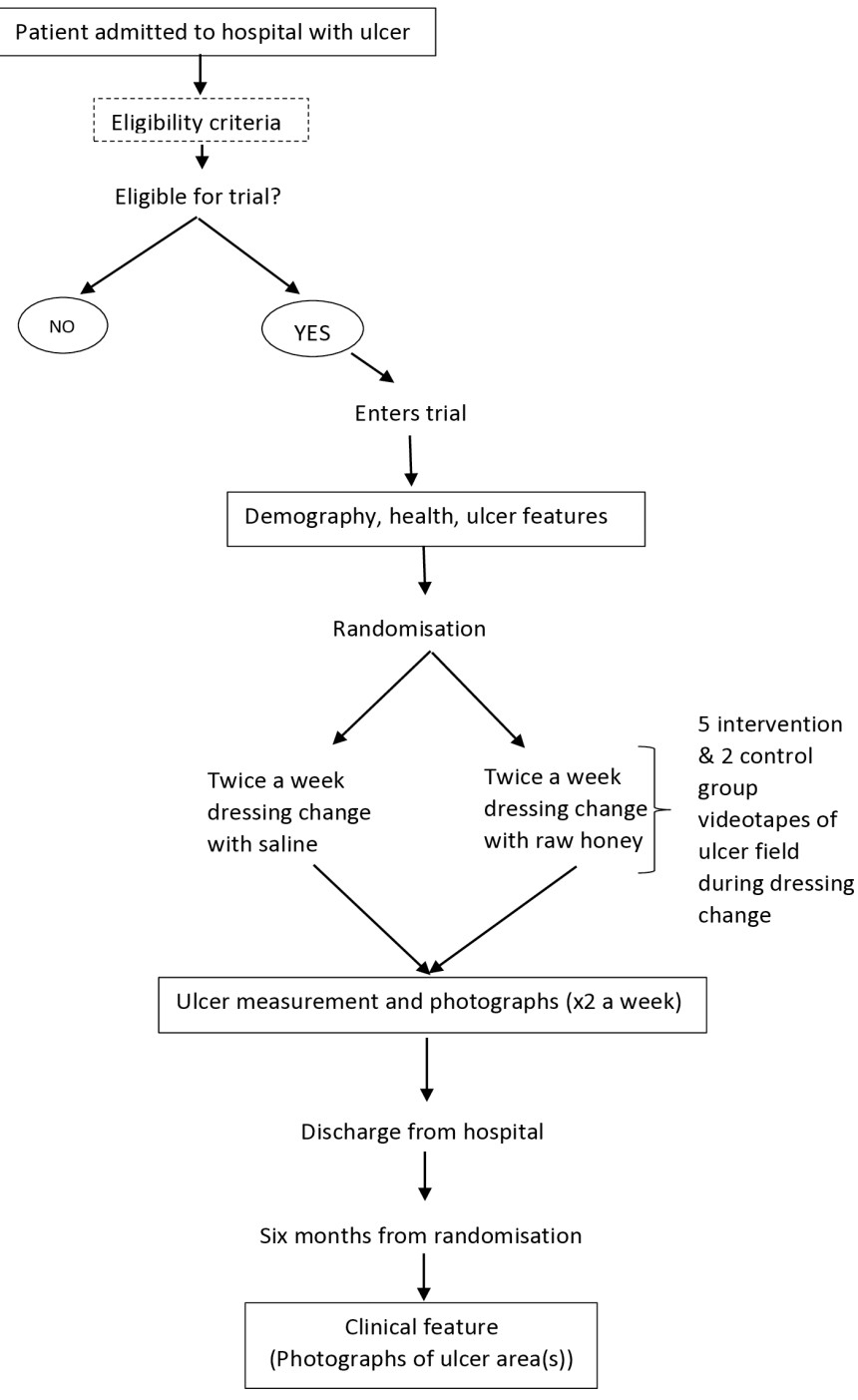

**Fig 1. The Pathway for recruitment of participants into the study.**

**Eligibility criteria (patients and ulcers)**

Eligible participants for inclusion in the study met the following criteria:

  i.  Patients with a chronic foot ulcer of at least 6 weeks duration due to leprosy neuropathy.

 ii.  ≥ 18 years of age.

iii.  Ulcer surface area between 2 and 20 cm$^2$.

 iv.  Ulcer is clean, dry, and free from infection.

  v.  The patients who can/and have signed the informed consent forms.

Those that failed to meet the inclusion criteria were excluded from participation in the study.

In scenarios where a participant has more than one foot ulcer, the largest ulcer that falls between 2 and 20 cm$^2$ was selected as the index ulcer before randomisation, and all other ulcers were treated in the standard way.

**Informed consent**

Researchers at the study centres were trained on Good Clinical Practice (GCP) and were responsible for obtaining consent from the study participants. They also screened the participants for eligibility before obtaining written informed consent. The participants' information sheet was translated into the local language (Hausa). All relevant information for the research participants in the study was contained in the participants' information sheet. The information sheet was handed to the participants a day before enrolment into the study. A copy of the information sheet used and the consent form are included as supplementary materials (S1 and S2 Texts).

**The intervention**

The intervention used for this study was raw, unadulterated honey obtained from local bee farmers in Nigeria. The honey samples were examined at the National Institute for Pharmaceutical Research and Development (NIPRD), Abuja, and confirmed to be free from Microbial contaminants. The honey was applied topically to the wound during dressing under strict hygienic conditions using sterilised equipment. The treatment was twice-weekly dressings conducted by nurses or paramedics.

**The control**

The participants in the control group received the usual care of twice-weekly normal saline dressings only. The normal saline dressing is the standard ulcer dressing method currently in use in the hospitals where the trial took place.

**Tracking of steps**

In line with standard practice, participants were discouraged from bearing weight on the ulcer site since weight bearing and the level of activity of patients might affect the rate of healing of the ulcers. Participants were asked to wear pedometers on the non-affected foot, and the step counts were recorded at each dressing change from the first dressing change until 84 days from randomisation or discharge, whichever came first.

**Discharge**

The trial participants were discharged at 84 days after randomisation or when healing (complete re-epithelialization) occurred before the 84th day of hospitalisation. The date of discharge was noted along with the participants' contact details, address, and contact details of at least one family member.

### Follow-up

The participants were contacted six months after randomisation for follow-up, during which the treated ulcer area was examined and photographed. The treated ulcer area is also examined for recurrence or the presence of any new ulcers.

### Sample size

A total of 130 participants were enrolled in the study. The sample size was chosen based on the two clinical outcomes: rate of healing and time to complete re-epithelialization. We anticipated that 70% of ulcers would heal within 84 days with standard care (according to a recent study of neuropathic ulcers, over half due to leprosy [19]). In that case, we assumed that the intervention would increase this proportion to 90% and hazards are constant and proportional (so that the hazard ratio is 1.91 for discharge), for a two-sided test of the hazard ratio with a type I error of 5% and statistical power of 80% and a 1:1 allocation ratio, 47 individuals are required in each group. With 130 participants, this allows for a dropout rate of up to 40% to achieve the power of 80%. At the most pessimistic sample size of 90, with a dropout rate of 40%, our minimum detectable effect size (i.e., the effect size that achieves an 80% power with 33 patients per arm) is a hazard ratio of 2.15, or 92.5% of patients in the treatment group being discharged by the end of the trial period. At the most optimistic sample size of 130 with no dropout, our minimum detectable effect size is a hazard ratio of 1.74, or 87.3% of patients being discharged in the treatment group. All calculations are based on a log-rank test.

We take a conservative approach and base the sample size calculation on the healing outcome and model with lower efficiency to ensure an adequate sample size for all main outcomes, since our inferential approach is not based on statistical significance but on a consideration of effect sizes and a comparison and triangulation of the totality of evidence. We were not concerned particularly with being "overpowered" or with a problem of multiple comparison (we expect the outcomes to be correlated), but with ensuring a sufficient sample size to estimate clinical effectiveness to a reasonable degree of precision [20,21].

### Randomisation and blinding

The participants were enrolled sequentially, stratified by ulcer size (above or below 10 cm$^2$), and randomly allocated (1:1) to undergo honey treatment or usual care with normal saline using a "digital sealed envelope" method [22,23].

An allocation table was generated remotely at the trials office at the University of Ibadan. A permuted block randomisation method was used to generate the randomisation sequence within each stratum. Randomly selecting blocks of size 2, 4, 6, or 8 was generated in order to maintain balance between the numbers allocated to each of the two groups and to ensure allocation concealment. The generated table was uploaded to the REDCap database (https://itm-redcap.bham.ac.uk/) and used for participant enrolment. Access to the allocation table was restricted, and staff in recruiting sites were not given access to the allocation table. When a participant's details were submitted, the trial arm and a unique study number were assigned and revealed to the local clinician so that the randomised group that the participant is assigned to could not be altered.

Only the overall research supervisor on site, the database managers in Birmingham, and the University of Ibadan trials unit, the clinical staff carrying out dressing changes in the room designated for this purpose, and participants themselves were aware of participants' randomly assigned group. The assessors were blinded to the treatment provided for randomly assigned groups.

### Data collection

As stated under 'eligibility criteria', patients were selected on the basis of an ulcer of 2–20 cm. We obtained standardised photographs of the index ulcers (see eligibility criteria) twice weekly during dressing changes for all participants [23]. The photographs were taken using the built-in camera in the tablet devices (Samsung Galaxy Tab S7, 13MP). The

photographs were taken perpendicular to the ulcer. For calibration purposes, a 3 cm size clean paper ruler with date and participant's trial identification number was placed in the photograph frame below the ulcer, but at the level of the skin. The photographs were then uploaded into the REDCap database and accessed by database managers at the University of Birmingham. The ulcer photos were then randomised and sent to the blinded assessors in Nepal for measurement of ulcer dimensions. The photographs were evaluated digitally by the designated observer in Nepal using the PictZar Digital Planimetry Software with an electronic PUSH Tool (National Pressure Injury Advisory Panel (NPIAP) at https://npiap.com/page/PUSHTool). This tool is a validated method for wound measurement with high intra- and inter-observer reliability (ICC > 0.95) [24–26]. The observers delineated an area of interest by manually 'painting' the ulcer area with colour using a computer mouse. The software then calculated the ulcer dimensions based on this profile.

We collected data on age, gender, height, weight (and hence Body Mass Index), the anatomical position of the ulcer, a composite test of neurological (motion and sensory) functioning – the Voluntary Muscle Testing and Semmes-Weinstein monofilament testing (VMT/SWMT), which is dichotomised into normal or impaired.

## Data management

All data generated from this study are classified according to the University of Birmingham Information Security Framework. All data were collected and stored electronically to reduce data entry errors, such as contradicting answers. Data was reported on an electronic Case Report Form (eCRF), and all local staff were trained to collect data directly onto electronic tablets. The data was stored on the REDCap database platform with access restricted by passwords at both the University of Ibadan and the local site in Nigeria. Each participant was allocated a unique study number when they agreed to participate (and before randomisation), which was used on all documents. A master list linking a trial participant number to their identity (name) was retained by the hospital securely in a locked filing cabinet.

## Statistical analysis

The primary analysis is the adjusted Cox proportional hazards model for time to complete healing (our key primary outcome), as this accounts for prognostic baseline factors (ulcer size and age) per standard practice in survival analyses for wound healing trials. All analyses are conducted on an intention-to-treat (ITT) basis, including all randomised participants with available outcome data (right-censored for non-healers at 84 days), consistent with CONSORT guidelines and the trial protocol. Unadjusted models and the linear mixed-effects analysis for healing rate are presented as supportive/secondary, with no post-hoc adjustments beyond the pre-specified multiple-testing correction (stepdown procedure on the two primary outcomes). For the rate of healing, we define the outcome ulcer size in $cm^2$ at each time point and include in the model time since admission, treatment status, and their interaction. We analysed this model using a linear mixed-effects model with participant-level random effects and both with and without adjustment for participant characteristics. Given there are multiple clinical outcomes (two outcomes, with and without adjustment), we adjusted reported p-values for multiple testing using a stepdown method, which provides an efficient means of controlling the family-wise error rate [22]. We also derived the approximate distributions of the test statistics to perform the stepdown procedure using a permutation test approach, by simulating 10,000 re-randomisations of the individuals [23].

We also compared the average daily step count between treatment and control groups as a simple difference in means (t-test). Since one group may stay longer in hospital than the other and since there may be an interaction between rate of healing and step count, we compared step counts over periods pre-set at 7, 14 and 42 days.

An interim analysis was conducted when 49 (three eights of the target) were followed up after discharge. The rationale for this analysis was for the detection of a 'penicillin-like' benefit or statistically significant negative effect of the treatment on either primary outcome. A statistical threshold of 0.01, one-sided (0.02 two-sided) was used for either primary outcome. Following the outcome of the interim analysis, the Independent Data Monitoring Committee (IDMC) examined the report and advised the Trial Steering Committee accordingly.

### Secondary treatment outcomes

For the outcomes recorded on the 6-month post-randomisation follow-up questionnaire, their analysis includes participants who provided their responses within 1 month (+/-) of the 6-month post-randomisation time point.

To examine the possible impact of the excluded data on the results, sensitivity analyses were performed on the 6-month follow-up outcome measures by including all participants.

The secondary outcomes are:

1. Recurrence of treated ulcer at 6 months from randomisation.

2. Appearance of a new ulcer at 6 months from randomisation.

3. Anatomical changes in the limb at 6 months from randomisation.

### Oversight and monitoring

The Leprosy Mission Nigeria is the study sponsor and oversees the study process in Nigeria. The University of Ibadan provided data management services and performed quality checks, while the University of Birmingham housed the data securely and performed quality checks.

The Trial Management Group (TMG) includes individuals at the University of Birmingham, The Leprosy Hospital Chanchaga, The Leprosy Mission Nigeria, RedAid Nigeria and the University of Ibadan Clinical Trials Unit who are responsible for the day-to-day management of the trial. The TMG met monthly by teleconference.

The Trial Steering Committee (TSC) provided the overall supervision of the trial and ensured that it was being conducted in accordance with the principles of Good Clinical Practice and other relevant regulations. The TSC kept oversight of the trial.

### Adverse event reporting and harms

The principal investigator in Nigeria is responsible for recording all Adverse Events (AEs) and reporting any Serious Adverse Events (SAEs) to the University of Birmingham and the University of Ibadan within 24 hours of the research staff becoming aware of the event. An SAE Form was made available on the tablets used to collect data and maintain a database of all safety/adverse events. The forms were reviewed by the TMG, which meets monthly and if required, also by the Project Manager. The TSG will periodically review all safety data and liaise with the Independent Data Monitoring Committee regarding any safety issues.

Any deaths were reported to the Sponsor irrespective of whether the death was related to the disease progression, the intervention or an unrelated event. Only deaths that could plausibly be caused by the intervention were reported to the Sponsor immediately.

## Results

### Screening and recruitment of participants

A CONSORT (CONsolidated Standards of Reporting Trials) flow chart showing participant recruitment and progression through the trial is presented in Fig 2. A total of 146 individuals were screened, of whom 130 met the inclusion criteria and were randomised. Sixteen [16] participants were excluded due to not meeting eligibility requirements. The 130 randomised participants were equally allocated to receive either normal saline dressing (n = 65) or honey dressing (n = 65). Throughout the study, 47 participants withdrew or were lost to follow-up at various stages.

At six months post-randomisation, follow-up was completed for 40 participants in the normal saline group and 43 participants in the honey group.

Three participants died after discharge; however, none of the deaths were related to the intervention or study procedures.

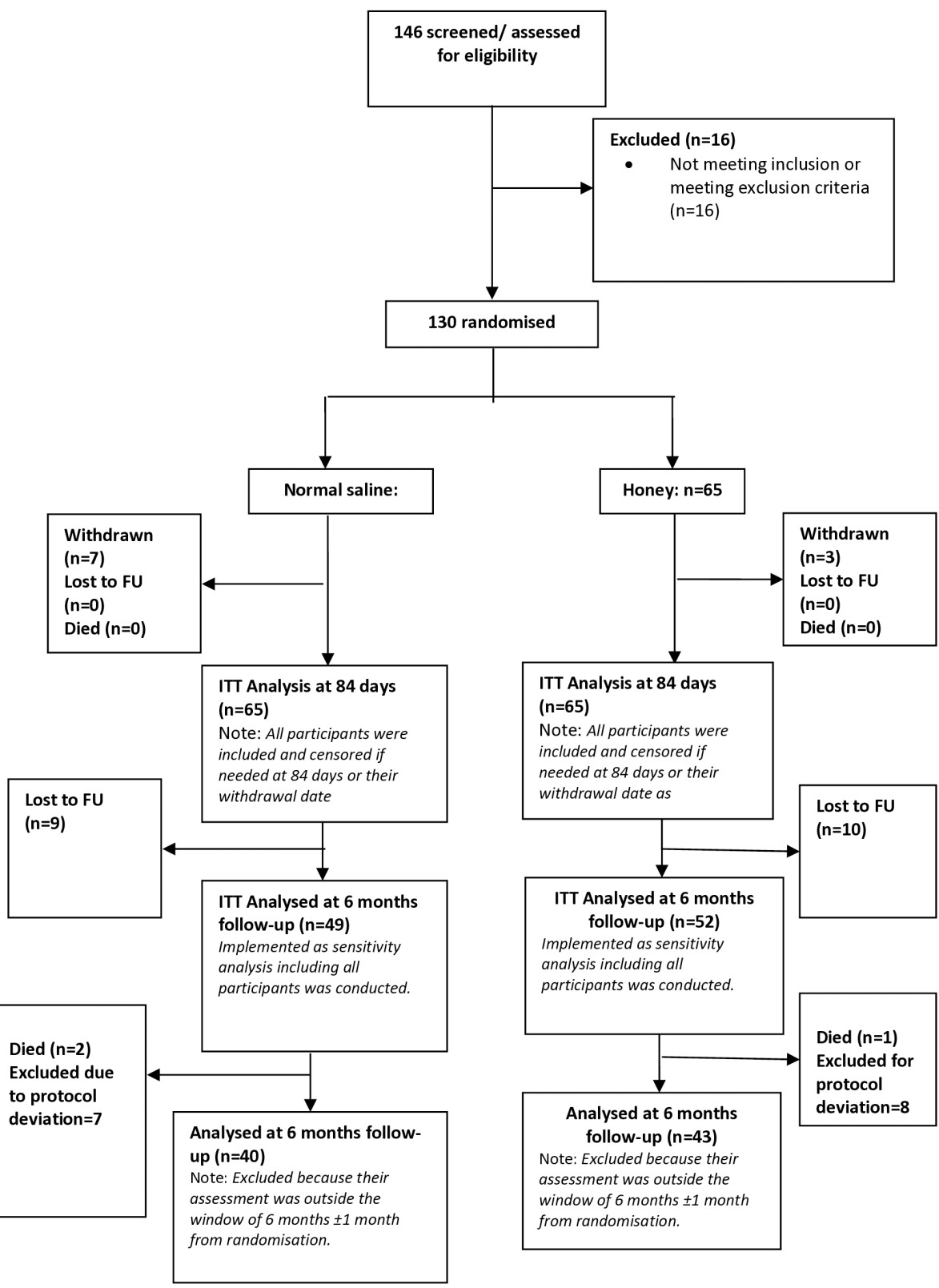

**Fig 2. CONSORT Flow diagram of the study.** FU = follow-up.

## Baseline characteristics

The mean age of participants at recruitment was 50.5 (SD = 15.1) years, with an age range of 18–82 years. The average size of the 130 ulcers at baseline was 9.1 cm² (SD = 9.7) with a median of 6 cm² (IQR, 4, 9). A total of 85 males and 45 females were enrolled in the trial. Most participants had no formal education, and only one had attained university-level education. The mean height, weight, and body mass index (BMI) were 162.8 (SD = 8.5) cm, 55.6(SD = 11.9) kg, and 21.0 (SD = 4.2), respectively. On average, participants had been living with leprosy for 21.3 (SD = 16.2) years. The trial ulcers had remained unhealed for a mean duration of 97.1 weeks prior to enrolment.

A detailed summary of baseline characteristics is presented in Table 1 below.

## Treatment outcomes

Two main outcome measures were evaluated: (1) the rate of wound healing ('primary' outcome), determined by the change in ulcer surface area per unit time based on biweekly measurements (cm²/time), and (2) the duration required for complete re-epithelialization, assessed at the 84-day mark.

**Table 1. Participant baseline characteristics.**

| | | Dressing changes with Normal saline (n = 65) | Dressing changes with Honey (n = 65) | Total (n = 130) |
|---|---|---|---|---|
| **Participant demographics** | | | | |
| Gender, n (%) | Male | 42 (64.6) | 43 (66.2) | 85 (65.4) |
| | Female | 23 (35.6) | 22 (33.9) | 45 (34.6) |
| Highest level of education | Never joined formal school | 37 (56.9) | 40 (61.5) | 77 (59.2) |
| | Primary level | 17 (26.2) | 16 (24.6) | 33 (25.4) |
| | Secondary level | 6 (9.2) | 4 (6.2) | 10 (7.7) |
| | Higher secondary level | 4 (6.2) | 5 (7.7) | 9 (6.9) |
| | University level | 1 (1.5) | 0 (0.0) | 1 (0.8) |
| **Clinical information** | | | | |
| Height in cm | Mean (SD) | 162.8 (8.9) | 162.9 (8.2) | 162.8 (8.5) |
| Weight in kg | Mean (SD) | 55.4 (11.5) | 55.8 (12.4) | 55.6 (11.9) |
| BMI | Mean (SD) | 20.9 (4.3) | 21.0 (4.2) | 21.0 (4.2) |
| **VMT/ST** | | | | |
| VMT/ST | Normal | 7 (10.8) | 3 (4.6) | 10 (7.7) |
| | Impaired | 58 (89.2) | 62 (95.4) | 120 (92.3) |
| **Current Ulcer Information** | | | | |
| Location of the trial ulcer | Left forefoot | 19 (29.2) | 25 (38.5) | 44 (33.9) |
| | Left midfoot | 26 (40.0) | 15 (23.1) | 41 (31.5) |
| | Left hindfoot | 7 (10.8) | 9 (13.9) | 16 (12.3) |
| | Right forefoot | 7 (10.8) | 5 (7.7) | 12 (9.2) |
| | Right midfoot | 4 (6.2) | 6 (9.2) | 10 (7.7) |
| | Right hindfoot | 2 (3.1) | 5 (7.7) | 7 (5.4) |
| **Measurements of the trial ulcer (for eligibility)ᵃ** | | | | |
| Trial ulcer Area (cm²) | Mean (SD) | 8.7 (8.3) | 9.6 (11.1) | 9.1 (9.7) |
| | Range | 2, 49 | 2, 63 | 2, 6 |

Data are either mean (SD) or number (%).

Fig 3 illustrates a logarithmic trend of ulcer size reduction over time, comparing the efficacy of honey-based dressings with that of normal saline dressings. Both interventions resulted in slow progressive reduction in ulcer size which was slightly faster in the honey group. However, the difference between the two treatments was not statistically significant.

The analysis of time to achieve healing or complete re-epithelialisation within 84 days post-randomisation is presented in Table 2. The unadjusted hazard ratio (HR) for honey compared to saline was 1.17 (95% CI: 0.60–2.27). The adjusted HR (for ulcer size and age) was 1.26 (95% CI: 0.64–2.47). In both cases, the confidence intervals included 1, indicating no statistically significant difference in healing rates between the two treatments but with a point estimate favoring the intervention.

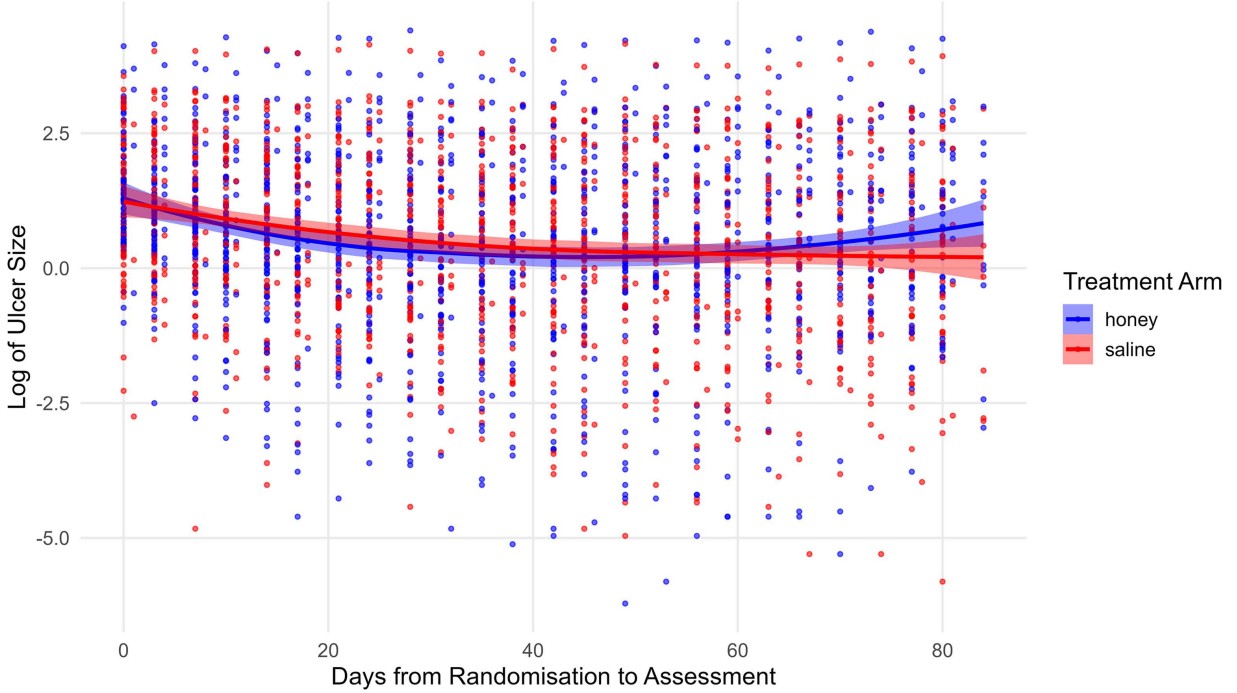

**Fig 3. Log of ulcer size over time by treatment arm.**

**Table 2. Analysis of primary outcome measure - Time to complete re-epithelisation.**

| Time point | | Dressing changes with Normal saline (n=65) | Dressing changes with Honey (n=65) | Unadjusted Hazard Ratio* (95% CI) | Adjusted‡ Hazard Ratio* (95% CI) |
|---|---|---|---|---|---|
| | Number of participants that withdrew before reached 84 days post randomisation | 5(7.7%) | 3 (4.6%) | | |
| At 84 days after randomisation | Number of Participants that had not reached complete re-epithelisation up to 84 days after randomisation | 44 (67.7%) | 43 (66.2%) | 1.17 (0.60, 2.27) | 1.26 (0.64, 2.47) |
| | Number of Participants that had complete re-epithelisation up to 84 days after randomisation | 16 (24.6%) | 19 (29.2%) | | |

‡Cox proportional hazards adjusted for the baseline values of trial ulcer size and patient age. Trial ulcer size and patient age was treated as continuous variables and considered as fixed effects in the adjustment. * HR > 1 means – hazard of 'healing' is higher in dressing changes with Honey.

## Test of proportional hazard assumption

The proportional hazards assumption was tested using Schoenfeld residuals. The global test (p = 0.28) and all covariates (age: p = 0.40; ulcer area: p = 0.65) met the PH assumption.

Other outcomes measured are included in the full analysis report attached as supplementary materials (S3 Text). The results for secondary outcomes are presented in Table 3 below, which showed that there was no difference between the arms.

## Step counts

Step counts were recorded using pedometers attached to each participant's foot to quantify activity levels. This method was used to examine the plausibility of 'performance bias' whereby patients in the intervention group may reduce weight bearing to a greater extent than patients in the control groups given that weight-bearing may impede the healing process. The analysis compared the number of steps taken by participants in the treatment group versus those in the control group. The results indicated that the difference in mean daily step counts between the two groups was not statistically significant. Step counts collected on days 7, 14, and 42 post-randomisation are presented in Table 4 and are very similar across groups.

## Discussion

The findings of this randomised controlled trial of raw honey dressings for treating chronic neuropathic ulcers in patients with leprosy demonstrated a modestly higher healing rate in the treatment group compared to the control group (29.2%

**Table 3. Analysis of binary secondary outcomes.**

| | | Dressing changes with normal saline (n = 49) | Dressing changes with Honey (n = 52) | Adjusted Relative Risk[1] (95% CI) | Adjusted Risk Differnce[2] (95% CI) |
|---|---|---|---|---|---|
| **Recurrence of treated ulcer at 6 months from randomisation** | | | | | |
| Yes | | 5 (10.2%) | 7 (13.5%) | 1.34 (0.46, 3.92) | 0.04 (-0.10, 0.19) |
| No | | 37 (75.5%) | 37 (71.2%) | | |
| Missing | | 0 | 0 | | |
| Excluded from analysis | | 7 (14.3%) | 8 (15.4%) | | |
| **Appearance of a new ulcer at 6 months from randomisation** | | | | | |
| Yes | | 2 (4.1%) | 1 (1.9%) | 0.33 (0.04, 2.93) | – |
| No | | 40 (81.6%) | 42 (80.8%) | | |
| Missing | | 0 | 1 | | |
| Excluded from analysis | | 7 | 8 | | |
| **Anatomical changes in the limb at 6 months from randomisation** | | | | | |
| 6 months | Normal | 7 (14.3%) | 2 (3.9%) | 1.14 (0.98, 1.32)[3] | |
| | Impaired | 35 (71.4%) | 42 (80.8%) | | |
| | Missing | 0 | 0 | | |
| | Excluded from analysis | 7 | 8 | | |

1: Log-binomial regression model adjusted for the baseline values of trial ulcer size and participant age. Trial ulcer size and participant age were treated as continuous variables and considered as fixed effects in this adjustment. Adjusted RR > 1 means a higher rate of recurrent or new ulcers, respectively, was observed in the dressing changes with the honey group.

2: Log-binomial regression model using the identity link function adjusted for the baseline values of trial ulcer size and participant age. Trial ulcer size and participant age were treated as continuous variables and considered as fixed effects in this adjustment. Adjusted RD > 0 means a higher risk of recurrent or new ulcers, respectively, for the dressing changes with the honey group.

3: Poisson regression model with robust standard errors adjusted for the baseline values of trial ulcer size and participant age. Trial ulcer size and participant age were treated as continuous variables and considered as fixed effects in this adjustment. Adjusted RR > 1 means a higher rate of normal limb was observed in the dressing changes in the honey group.

**Table 4. Analyses of activity measurement (step count) at 7, 14, and 42 days post-randomisation.**

| Activity measurement | | Dressing changes with normal saline (N = 65) | Dressing changes with Honey (N = 65) | Mean Difference[1] 95% CI |
|---|---|---|---|---|
| Average daily steps measured at 7 days post randomisation | N | 61 | 61 | 79.9 (-467.1, 626.9) |
| | Mean (SD) | 1859.6 (1628.4) | 1779.7 (1415.5) | |
| | Min - Max | 0 – 6690.3 | 0 – 6263.4 | |
| | Missing | 4 | 4 | |
| Average daily steps measured at 14 days post randomisation | N | 56 | 61 | 39.0 (-540.4, 618.4) |
| | Mean (SD) | 2039.5 (1678.8) | 2000 (1573.9) | |
| | Min - Max | 0 - 6993 | 0 - 7002 | |
| | Missing | 9 | 4 | |
| Average daily steps measured at 42 days post randomisation | N | 49 | 49 | 67.7 (-714.8, 850.3) |
| | Mean (SD) | 2381.6 (1817.3) | 2313.9 (2076.8) | |
| | Min - Max | 0 – 6504.1 | 0 – 12388.9 | |
| | Missing | 16 | 16 | |

1: Mean difference is estimated using a t-test. MD > 0 indicates higher mean daily steps in dressing changes with honey group.

vs. 24.6% complete re-epithelialisation at 84 days), the difference was not statistically significant (adjusted HR: 1.26, 95% CI: 0.64–2.47). Similarly, ulcer recurrence and new ulcer development at six months post-randomisation were comparable between the two groups. These results suggest that while honey may offer some benefits, its superiority over normal saline dressings remains inconclusive within the context of this study.

The observed trend towards improved healing with honey aligns with historical and contemporary evidence supporting honey's wound-healing properties. Honey has been shown to promote tissue growth, facilitate debridement, and exhibit antimicrobial effects in various wound types, as highlighted by Jull et al. [14] and Subrahmanyam and Ugane [17]. However, the lack of statistical significance in our findings contrasts with some studies reporting more pronounced benefits of honey in chronic wounds, such as diabetic foot ulcers and venous leg ulcers [14]. This discrepancy may stem from differences in wound aetiology, as leprosy-related ulcers are uniquely complicated by neuropathy and reduced blood supply, factors that may attenuate the therapeutic effects of honey [1,6].

The study's results also resonate with the broader literature on leprosy ulcer management, which underscores the challenges of achieving sustained healing in neuropathic wounds. For instance, Napit et al. [6] emphasised the recalcitrant nature of leprosy ulcers and the need for innovative treatments, though their trial focused on autologous blood products rather than honey. Similarly, Reinar et al. [7] noted the paucity of high-quality evidence for ulcer treatments in leprosy, a gap this study sought to address. Our findings contribute to this limited evidence base but highlight the need for further research to clarify honey's role.

The study's rigorous design, prospective, randomised, and single-blinded, strengthens its internal validity. The use of standardised photographic assessment and blinded evaluators in Nepal minimised measurement bias, a common limitation in wound-healing trials [24]. However, the high dropout rate (36.2%) and relatively small sample size may have limited the study's power to detect significant differences. The sample size calculation assumed a 70% healing rate in the control group, but the observed rate was lower (24.6%), potentially reflecting the refractory nature of leprosy ulcers in this population. Future studies with larger cohorts and longer follow-up periods are warranted to confirm or refute these findings.

Despite the non-significant results, the trend favouring honey dressings suggests that this intervention may still hold promise, particularly in resource-limited settings where honey is readily available and cost-effective. The safety profile of honey, with no reported adverse events in this trial, further supports its consideration as an adjunct therapy.

However, clinicians should weigh these potential benefits against the lack of definitive evidence and consider multimodal approaches, including offloading and self-care practices, as recommended by WHO guidelines [8,9,27].

The study's limitations include its single-blinded design, which may have introduced performance bias, and the heterogeneity of ulcer characteristics at baseline. However, the step counts were similar across groups and any performance bias was clearly insufficient to yield a positive result. The follow-up period of six months may have been insufficient to capture long-term recurrence rates, but the hypothesis was related to healing, not recurrence. The use of local honey introduces variability in composition that could influence outcomes, a problem for all non-pharmacologically prepared formulations in wound care.

In summary, this trial adds to the sparse literature on leprosy ulcer management but does not provide conclusive evidence for the superiority of honey over normal saline dressings. The findings underscore the complexity of treating neuropathic ulcers and the need for further high-quality studies with larger, more diverse populations to explore honey's potential role. Until such evidence emerges, honey may be considered a safe and feasible option, though not a definitive solution, for leprosy-related ulcer care.

## Conclusion

This randomised controlled trial represents one of the first robust evaluations of honey dressings for chronic neuropathic ulcers in leprosy patients. While the results demonstrated a modest, non-significant trend towards improved healing with honey compared to standard saline dressings, they ultimately failed to show conclusive superiority of this traditional therapy. The findings contribute valuable data to the limited evidence base for leprosy ulcer management.

The study's methodological strengths, including its randomised design, blinded outcome assessment, and standardised intervention protocol, provide a solid foundation for interpreting these results. However, the high attrition rate and relatively small sample size underscore the need for larger, multicentre trials with longer follow-up periods. Future research should explore optimal honey formulations, potential synergistic effects with other therapies, and strategies to enhance treatment adherence in this vulnerable population.

From a clinical perspective, we recommend a comprehensive wound care approach in resource-limited settings where leprosy remains endemic. However, we also reinforce the importance of holistic management strategies that address the multifactorial nature of neuropathic ulcers, including pressure offloading, infection control, and patient education. The continued search for more effective interventions for leprosy-related ulcers remains a critical priority in global neglected tropical disease research. We would urge any clinicians who wish to use honey treatment offer patients entry into randomised trials with blinded outcome assessment.

## Supporting information

**S1 Text. Trial information sheet.** Contains details of the information sheet that was handed out and explained to the participants to decide to voluntarily participate in the trial.
(DOCX)

**S2 Text. Trial consent form.** The detailed written informed consent document that was signed by all the participants before enrollment into the trial.
(DOCX)

**S3 Text. Analysis final report.** The final report prepared by the trial statisticians.
(DOCX)

**S1 CONSORT Checklist. CONSORT checklist.** Checklist of items in accordance with the CONSORT guidelines (Adapted from https://journals.lww.com/asnjournals/Documents/CONSORT%202025.pdf).
(PDF)

## Acknowledgments

We gratefully acknowledge the many individuals whose contributions made this study possible. Foremost, we extend our sincere appreciation to the persons affected by leprosy in Nigeria, whose support and cooperation were fundamental to the success of this work. We are also indebted to the management and staff of the two hospitals where the study was conducted, The Leprosy Referral Hospital, Chanchaga, and St Benedict Tuberculosis and Leprosy Relief Hospital, Ogoja, for their invaluable assistance. Furthermore, we thank the professionals who served on the independent trial and monitoring teams, as well as the blind assessors, for their expert input and relentless support.

## Author contributions

**Conceptualization:** Paul Alumbugu Tsaku, Sunday Odihiri Udo, Pius Sunday, Sopna Choudhury, Richard Lilford.

**Data curation:** Abiola Oladejo, Joshua Akinyemi, Akinyinka Omigbodun, Sopna Choudhury, Samuel Watson.

**Formal analysis:** Joshua Akinyemi, Samuel Watson, Richard Lilford.

**Funding acquisition:** Richard Lilford.

**Investigation:** Paul Alumbugu Tsaku, Anthony Meka, Linda Chinonso Ugwu.

**Methodology:** Paul Alumbugu Tsaku, Sunday Odihiri Udo, Pius Sunday, Anthony Meka, Linda Chinonso Ugwu, Abiola Oladejo, Joshua Akinyemi, Akinyinka Omigbodun, Sopna Choudhury, Jo Sartori, Onaedo Ilozumba, Samuel Watson, Richard Lilford.

**Project administration:** Paul Alumbugu Tsaku, Sunday Odihiri Udo, Pius Sunday, Anthony Meka, Linda Chinonso Ugwu, Abiola Oladejo, Joshua Akinyemi, Akinyinka Omigbodun, Sopna Choudhury, Jo Sartori, Onaedo Ilozumba, Samuel Watson, Richard Lilford.

**Resources:** Sunday Odihiri Udo, Pius Sunday, Sopna Choudhury, Jo Sartori, Onaedo Ilozumba, Richard Lilford.

**Software:** Joshua Akinyemi, Sopna Choudhury, Samuel Watson, Richard Lilford.

**Supervision:** Sunday Odihiri Udo, Pius Sunday, Akinyinka Omigbodun, Sopna Choudhury, Jo Sartori, Samuel Watson, Richard Lilford.

**Validation:** Abiola Oladejo, Joshua Akinyemi, Sopna Choudhury, Samuel Watson, Richard Lilford.

**Visualization:** Abiola Oladejo, Sopna Choudhury.

**Writing – original draft:** Paul Alumbugu Tsaku, Sopna Choudhury, Richard Lilford.

**Writing – review & editing:** Paul Alumbugu Tsaku, Sunday Odihiri Udo, Pius Sunday, Anthony Meka, Linda Chinonso Ugwu, Abiola Oladejo, Joshua Akinyemi, Akinyinka Omigbodun, Sopna Choudhury, Jo Sartori, Onaedo Ilozumba, Samuel Watson, Richard Lilford.

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
