## [Decision Letter · Decision Letter 0]

16 Oct 2025

A randomised controlled trial of raw honey for the healing of ulcers in leprosy in Nigeria

Dear Dr. Tsaku,

Thank you for submitting your manuscript to PLOS Neglected Tropical Diseases. After careful consideration, we feel that it has merit but does not fully meet PLOS Neglected Tropical Diseases's publication criteria as it currently stands. Therefore, we invite you to submit a revised version of the manuscript that addresses the points raised during the review process.

Please submit your revised manuscript within 60 days. If you will need more time than this to complete your revisions, please reply to this message or contact the journal office at plosntds@plos.org. Please include the following items when submitting your revised manuscript:

We look forward to receiving your revised manuscript.

Kind regards,

Peter Steinmann, Ph.D.

Academic Editor

Ana LTO Nascimento

Section Editor

Shaden Kamhawi

co-Editor-in-Chief

Paul Brindley

co-Editor-in-Chief

**Additional Editor Comments :**

Please carefully consider and respond to all comments, with special consideration to reviewer #2!

**Journal Requirements:**

At this stage, the following Authors/Authors require contributions: Paul Alumbugu Tsaku, Sunday Odihiri Udo, Pius Ogbu Sunday, Anthony Meka, Linda Chinonso Ugwu, Abiola Oladejo, Joshua Akinyemi, Akinyinka Omigbodun, Sopna Choudhury, Jo Sartori, Onaedo Ilozumba, Samuel Watson, and Richard Lilford. Please ensure that the full contributions of each author are acknowledged in the "Add/Edit/Remove Authors" section of our submission form.

- TM on page: 10.

5) We have noticed that you referred to supplementary materials on page 22. However, there are no corresponding files uploaded to the submission. Please upload them as separate files with the item type 'Supporting Information'.

**Reviewers' Comments:**

Reviewer's Responses to Questions

**Key Review Criteria Required for Acceptance?**

**Methods**

-Are the objectives of the study clearly articulated with a clear testable hypothesis stated?

-Is the study design appropriate to address the stated objectives?

-Is the population clearly described and appropriate for the hypothesis being tested?

-Is the sample size sufficient to ensure adequate power to address the hypothesis being tested?

-Were correct statistical analysis used to support conclusions?

-Are there concerns about ethical or regulatory requirements being met?

Reviewer #1: Table 1: one of the characteristics is VMT/ST - this is the name of a test that includes many examinations, so what does normal and impaired VMT/ST mean in this context? The is a separate line for loss of sensation - so that is the ST part of 'VMT/ST. We then have 'loss of motor function' - what motor function is being tested in this context? is it only dorsi-flexion, or is it more than that?

The 84 days is based on a study that used total contact casts - this study did not use TCCs, so not comparable.

As this is a research study, the honey was tested in a laboratory before being used, to ensure it was not contaminated with microbes. This would not be the case if used routinely. Do the authors have any advice or caution for using honey that has not been tested for contamination?

Reviewer #2: (No Response)

Reviewer #3: The study's objective to test the hypothesis that raw honey improves the healing rates of leprosy-related ulcers compared to standard saline is clearly stated and testable. The dual-centre, single-blinded RCT design is appropriate, and the inclusion and exclusion criteria are well-defined.

Sample size justification is reasonable, although the power is reduced due to high attrition. Statistical analyses used in the study are suitable and well interpreted. Ethical procedures are well described with all the necessary approvals and informed consent in place.

**Results**

-Does the analysis presented match the analysis plan?

-Are the results clearly and completely presented?

-Are the figures (Tables, Images) of sufficient quality for clarity?

Reviewer #1: (No Response)

Reviewer #2: (No Response)

Reviewer #3: The results are clearly presented and supported by appropriate tables. Figures, including the CONSORT flow diagram, are clear and informative. The primary outcome shows a modest but non-significant trend favouring honey. The secondary outcomes (recurrence, new ulcers) are presented clearly and support the main finding of no significant difference.

It could have been interesting to understand if any significant instructions on selfcare were provided post discharge, which had an influence on prognosis.

Figures are acceptable.

**Conclusions**

-Are the conclusions supported by the data presented?

-Are the limitations of analysis clearly described?

-Do the authors discuss how these data can be helpful to advance our understanding of the topic under study?

-Is public health relevance addressed?

Reviewer #1: Honey is stated as a safe, affordable option in resource-limited settings. But we are not given the cost compared to normal saline.

Reviewer #2: (No Response)

Reviewer #3: The conclusions are balanced and supported by the data. The authors appropriately highlight honey's potential safety and acceptability without overstating efficacy. Limitations and public health relevance are clearly discussed.

**Editorial and Data Presentation Modifications?**

Reviewer #1: none

Reviewer #2: (No Response)

Reviewer #3: No modifications needed. It can be accepted.

**Summary and General Comments**

Reviewer #1: The study repeats the importance of pressure relief for healing wounds, rather than the wound cleaning material/solution.

Reviewer #2: Introduction:

Para 1 - is it the pathogen itself or the immune response to that pathogen that causes nerve damage? Or both?

Para 2- in the list of other things people have tried I think it should be made clearer that the evidence for benefit for ost of these interventions is very minimal.

Methods

The formatting of the flow diagram is poor with words spilling out of the boxes etc. Please try and correct this.

Consent is not fully described - was thias written consent?

Why were the specific study sites chosen?

Please make clear in the methods that the trial was pre-registered (I can see this is given elsewhere but should also ideally be in the methods).

Please review the CONSORT checklist for RCTs and ensure that the paper is reported in line with this. Please submit a CONSORT checklist with the paper.

Please describe the intervention in more details. For example does all the honey come from a single producer? Is there inter-batch variation? What fabric material was used for both control and intervention dressings?

Is the approach taken to measuring ulcer size using photos and the software validated? Is there good reproducibility? And inter-observer agreement - it seems critical some evidence for this is provided as it serves as the key outcome measure.

I am unclear if the primary outcome was the proportion healed at end of follow-up OR the time to healing? Was the protocol and the SAP published in advance?

What is the PRIMARY analysis - is it with or without adjustment. As currwently written the stats plan is very difficult to decide what was specified a priori and what was post-hoc. Is the primary analysis ITT or PP? This isnt stated.

When someone had >1 ulcer how was the outcome ascertained? Time to healing of a specific ulcer? Of all ulcers?

The trial was funded in the UK but there doesnt appear to be any ethics approval in the UK?

Throughout there are averages given without measures of dispersion (i.e the mean ulcer size was X but what was the SD, is a mean the best measure or a median?) I appreciate some of these are in tables but its good practice to include in the text too.

Tablr 1 is very long. I suspect this couldbe summarised with more detailed information in a supplementary table.

In the results please avoid using language that suggests a positive difference when in fact there was no statistically significant differences found. for example ' " the slope of reduction was steeper in the honey-treated group, suggesting a more favourable healing trajectory. " O" With the unadjusted hazard ratio (HR) of 1.17 (95% CI: 0.60–2.27), honey had a 17% higher chance of healing compared to saline, but the confidence interval includes 1 (no effect), so this is not statistically significant." Please review all of the results thoroughly for such language.

Figure 3 is incredibly hard to intrepet. Please consider a clearer visualisation.

Table 3 seems to be raw Stata output - is this really needed?

Dont describe sensitivity analysis methods in the results. These should be in the methods and it should be clear if these were pre-specified or post-hoc.

Reviewer #3: The study is a high quality clinical trial, which is well designed, ethically sound and focuses on addressing the crucial component of ulcer management in Leprosy. Despite the results being neutral, the study contributes valuable evidence and demonstrates high research quality in the NTD settings.

PLOS authors have the option to publish the peer review history of their article (what does this mean? ). If published, this will include your full peer review and any attached files.

**Do you want your identity to be public for this peer review?** For information about this choice, including consent withdrawal, please see our Privacy Policy .

Reviewer #1: No

Reviewer #2: No

Reviewer #3: No

**Figure resubmission:**
---

## [Editor Report · Decision Letter 1]

17 Dec 2025

Dear Dr. Tsaku,

We are pleased to inform you that your manuscript 'A randomised controlled trial of raw honey for the healing of ulcers in leprosy in Nigeria' has been provisionally accepted for publication in PLOS Neglected Tropical Diseases.

Best regards,

Peter Steinmann, Ph.D.

Academic Editor

Ana LTO Nascimento

Section Editor

Shaden Kamhawi

co-Editor-in-Chief

Paul Brindley

co-Editor-in-Chief

---

## [Editor Report · Acceptance letter]

Dear Dr Tsaku,

We are delighted to inform you that your manuscript, "A randomised controlled trial of raw honey for the healing of ulcers in leprosy in Nigeria," has been formally accepted for publication in PLOS Neglected Tropical Diseases.

Best regards,

Shaden Kamhawi

co-Editor-in-Chief

Paul Brindley

co-Editor-in-Chief
